# KEPL: Knowledge Enhanced Prompt Learning for Chinese Hypernym-Hyponym Extraction

**Ningchen Ma[1,3]\*, Dong Wang[2]\*, Hongyun Bao[3]†, Lei He[2], Suncong Zheng[2]**
[1]China University of Mining & Technology, Beijing
[2]Tencent Machine Learning Platform, Tencent, Beijing, China
[3]Institute of Automation, Chinese Academy of Sciences, Beijing, China
maningchen0901@gmail.com, hongyun.bao@ia.ac.cn
{thudongwang,bettyleihe,congzheng}@tencent.com

## Abstract

Modeling hypernym-hyponym ("is-a") relations is important for many natural language processing (NLP) tasks, such as classification, natural language inference and relation extraction. Existing work on is-a relation extraction is mostly in the English language environment. Due to the flexibility of language expression and the lack of high-quality Chinese annotation datasets, it is still a challenge to accurately identify such relations from Chinese unstructured texts. To tackle this problem, we propose a Knowledge Enhanced Prompt Learning (KEPL) method for Chinese hypernym-hyponym relation extraction. Our model uses the Hearst-like patterns as the prior knowledge. By exploiting a Dynamic Adaptor to select the matching pattern for the text into the prompt, our method simultaneously embedding patterns and text. Additionally, we construct a Chinese hypernym-hyponym relation extraction dataset, which contains three typical scenarios, as Baidu Encyclopedia, news and We-media. The experimental results on the dataset demonstrate the efficiency and effectiveness of our proposed model.

## 1 Introduction

Hypernym Discovery is a core work in taxonomy construction (Wang et al., 2019). Hypernym relation is a semantic relation that exists between a term (hyponym) and a more general or abstract term (hypernym). Due to its capacity for representing semantic relations, hypernym becomes an essential concept in modern natural-language research, and is a fundamental component in many natural language processing (NLP) tasks, such as question answering (Yang et al., 2017; Yu et al., 2021), taxonomy construction (Chen et al., 2019; Luo et al., 2020) and personalized recommendation (Huang et al., 2019).

---

\*These authors contributed equally to this work.
†Corresponding Author

Typical efforts in Hypernym Discovery can be roughly classified into two main types: rule-based methods and detection-based methods. Rule-based methods (Auger and Barrière, 2008; Kliegr et al., 2008; Seitner et al., 2016; Snow et al., 2004; Wang et al., 2017) rely on predefined linguistic rules or patterns to extract hyponym-hypernym relations. These rule-based methods capture specific syntactic or semantic structures that indicate a hierarchical association between terms, however, rule-based methods lack sufficient capability to discover the hyponym-hypernym relation embedded in the semantic content of sentences. For example, in the sentence 'Jay Chou, an acclaimed singer, mesmerizes audiences with his soulful performances.' the rule-based approach might fail to recognize the hyponym-hypernym relation between 'Jay Chou' and 'singer'.

Another effective way is to model this problem as a hypernym detection task (Dash et al., 2020a; Roller et al., 2018; Yamane et al., 2016). These methods handle the hypernym detection task in a pipelined manner, i.e., extracting the entities first and then recognizing hypernym relations. This separate framework makes each component more flexible, but it neglects the relevance between two sub-entities.

Identifying hyponym-hypernym relation by modeling the interaction of sparse attributes, many studies are dedicated to constructing English hypernym-hyponym relationship datasets (Berend et al., 2018; Bernier-Colborne and Barrière, 2018). Meanwhile, Chinese language has unique linguistic characteristics and categories that need to be considered (Zhang et al., 2022b). The lack of such datasets has hampered the progress of Chinese hyponymy extraction research. Even though Chinese speakers account for a quarter of the world population, there has been no existing high-quality dataset for Chinese hypernym relation extraction.

In this paper, we propose the Knowledge En-

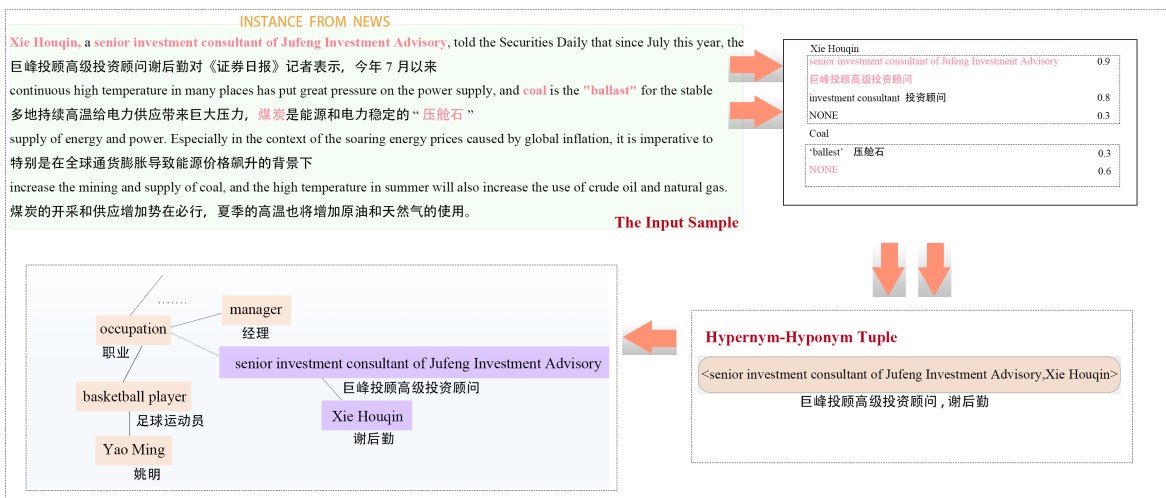

Figure 1: Hypernym discovery for taxonomy construction

hanced Prompt Learning (KEPL) method which utilizes both knowledge and latent semantics for end-to-end Chinese hypernym relation extraction. Specifically, our method employs a Dynamic Adaptor for Knowledge which can adaptively construct a unified representation for both the structured prior knowledge and the unstructured text context. To facilitate a more coherent integration of the structured prompts and unstructured text, we employ a mechanism that learns a unified representation of context through specific attention. For span selection, we use the focal loss to counteract the issue of sample imbalance, a commonly observed phenomenon in extractive tasks.

The lack of specific hypernym relation datasets also leads to deficiencies in current works (Chen et al., 2019; Luo et al., 2020). To address this challenge, we propose the CHR dataset that aims to improve the coverage and accuracy of hypernym relations in taxonomies. We believe that our dataset can contribute to the development of more accurate and comprehensive taxonomies.

The main contributions of our work are summarized as follows:

1. To the best of our knowledge, there currently exists no commonly used dataset for Chinese hypernym-hyponym discovery. We construct a Chinese hypernym relation extraction dataset, which contains three typical scenarios, as follows as Baike, news and We-media. The proposed dataset with multiple data sources can well cover specific expressions in various corpus.

2. We propose a novel framework, the Knowledge Enhanced Prompt Learning (KEPL) which leverages prior knowledge as prompts and transfers prior knowledge into an extraction task. Our framework learns a unified representation of context through specific attention, proving effective for various natural language processing tasks, including but not limited to taxonomy construction and semantic search.

3. Our extensive experiments on the proposed dataset have shown that the KEPL framework achieves a 2.3% improvement in F1 over the best method, demonstrating the effectiveness of our approach. Further, we show the results of individually removing components from the trained KEPL on CHR dataset and proved the effectiveness of each component.

## 2 Related Work

**Hypernym Dection** Research into hypernym relation extraction has mainly used unsupervised methods, falling into two categories: pattern-based and distributional approaches.

The pattern-based approach (Navigli and Velardi, 2010; Boella and Di Caro, 2013; Vyas and Carpuat, 2017; Bott et al., 2021), established by (Hearst, 1992; Wang and He, 2020), employs specific predefined linguistic patterns, such as 'is-a' and 'including', to detect hypernym relations. While this approach is simple and widely applicable, it is constrained by its reliance on predefined patterns, sensitivity to sentence structure, and the necessity for manual resource curation. Distributional approaches like (Fu et al., 2014) use a distant supervision method for extracting hypernyms from various sources. Their models produce a list

of hypernyms for a given entity. Subsequently, (Sanchez and Riedel, 2017) highlights the unique performance of the Baroni dataset in providing consistent results, attributing its effectiveness to its alignment with specific dimensions of hypernymy: generality and similarity

Additionally, hybrid methods (Bernier-Colborne and Barrière, 2018; Dash et al., 2020b; Yu et al., 2020) that amalgamate different techniques have been explored. For instance, (Held and Habash, 2019) proposed a method that merges hyperbolic embeddings with Hearst-like patterns, resulting in better performance on various benchmark datasets.

**Prompt learning** Prompt-based learning, a novel paradigm in pretrained language models (Zhang et al., 2022a), restructures downstream tasks to better align with pre-training tasks, enhancing the model's performance. A notable application of this approach is demonstrated by (Schick and Schütze, 2021), where classification problems are transformed into cloze tasks. This is achieved by creating relevant prompts with blanks and establishing a mapping from specific filled words to the corresponding predicted categories. This method effectively bridges the gap between the task and the model's training. Furthermore, (Ma et al., 2022) introduces a model named PAIE, which leverages prompts for Event Argument Extraction (EAE) at both sentence and document levels. This innovative use of prompts in EAE tasks showcases the versatility and efficiency of the prompt-based learning approach.

## 3 Data

We introduce the CHR (Chinese Hypernym Recognition) Dataset – an innovative resource that specifically addresses the current shortcomings in Chinese hypernym discovery. The key idea of constructing the CHR dataset is to enhance the quality and diversity across various domains, which are currently insufficient in existing resources.

### 3.1 Data source

To address the existing limitations in Chinese hypernym discovery, particularly the lack of diversity, we constructed the CHR dataset.

Our dataset is constructed by incorporating data from three distinct sources: encyclopedic knowledge, We-Media public accounts and news.

We gather the Baike data from a variety of reputable Baidu Baike online encyclopedias. However, we strategically omitted entries that were too short, lacked contextual richness, or had content outside the scope of our study, such as stub entries and disambiguation pages. The We-Media data was gathered from a wide range of accounts as lifestyle, entertainment, technology, and education. Our scraper was programmed to regularly check these accounts for the latest articles and retrieve historical articles where possible. Advertisements and unrelated links were excluded from our dataset; we focused only on preserving the main content of the articles.

Our dataset includes news data from multiple high-profile news platforms such as Xinhua News Agency, Tencent News, and Sina News. To eliminate domain bias, we selected articles from a broad spectrum of categories including politics, economics, sports, culture, and science, and we eliminated articles with insufficient text, duplicate articles, and those not pertaining to our research from the dataset.

### 3.2 Data construction

**Pre-processing** After gathering data from the various sources, we remove irrelevant features which contain HTML tags, special characters, and formatting. We also performed tokenization, converting sentences into tokens for further processing.

Sentences with a character count falling below the established threshold of 10 characters were disregarded. Those exhibiting an overuse of punctuation marks such as commas, colons, and periods were excised. Additionally, sentences embodying numerals or particular non-Chinese characters were filtered out.

The sentences that have passed through these filters are then manually labeled with the appropriate hypernym-hyponym relation by our team. We mark the entity position by determining the start $entity_{start}$ and end $entity_{end}$ positions of the answer in the text, aiming for accurate identification of the relation scope. If a sentence does not contain explicit hypernym-hyponym entities, the corresponding positions are marked as NONE. Following the above extraction, the potential pairs were presented to a team of trained individuals for review.

**Final Format**  We streamline and structure our data into a uniform and easily accessible format, facilitating subsequent analysis and modeling.

We use the data consisting of a set of (data, span1,span2), Which span1 represents the latent position of hypernym, span2 represents the latent position of the hyponym. An excerpt for one example: 链接检验器（link checker）是测试并报告网站的页面内的超文本链接有效性的程序. In this example, 程序 is the hypernym of the 链接检验器. The format of the output is set as < 程序, 链接检验器.>. We conducted a thorough examination of the data and implemented a meticulous annotation process to ensure the quality and reliability of the dataset, the detail can be seen in Table 1.

Table 1: The detail of the proposed CHR dataset

|       | Context | Hypernym |
| ----- | ------- | -------- |
| Train | 21376   | 12006    |
| Dev   | 2672    | 1431     |
| Test  | 2672    | 1475     |

## 4  Method

In this section, we describe our Knowledge Enhanced Prompt Learning (KEPL) framework, specifically designed to extract informative hypernym arguments from text documents. The overall structure of KEPL is depicted in Figure 2.

### 4.1  KEPL framework

**Preliminary**  We focus on the problem of hypernym discovery. Each document is represented as a set $doc = \{d_1, \ldots, d_N\}$ and is mapped to a set of spans, $S = \{S_1, S_2, \ldots, S_k\}$, using a specific prompt $Pm_i$. Here, each span consists of a pair $H_u, H_d$, where $H_d$ denotes a hyponym and $H_u$ represents a hypernym.

**KEPL Structure**  Our KEPL structure is based on a sequence-to-sequence pre-trained language model, primarily used for text generation. We first explain how we inject different knowledge into a given sentence (sec 4.2). Subsequently, we use an attention mechanism to learn a Unified Representation for Templates and Context, allowing us to capture valuable information from both structured prompts and unstructured text in Section 4.3. Lastly, we calculate the distribution of each token

being selected as the start or end of each role feature in Section 4.4.

### 4.2  Dynamic Adaptor for Knowledge

In the proposed KEPL model, our aim is to construct a unified representation for both the structured prior knowledge and the unstructured text context. To achieve this, we use the Hearst method which facilitates the identification of hypernym ("is-a") relations within a sentence by exploiting certain lexico-syntactic patterns. By introducing this prior knowledge in the form of lexico-syntactic patterns, we aim to increase the model's capability to precisely identify hypernym and hyponym relations.

Firstly, we represent each structured prior knowledge (the lexico-syntactic patterns) in the form of prompts, and use the corresponding prompt representations, $E_p$ as follows:

$$E_p = L_{dec}\left(pmi\right) \tag{1}$$

Here, $pmi$ represents a set of structured prompts with Hearst patterns, and $E_p$ denotes the corresponding embeddings obtained from the decoder module $L_{dec}$ of the pre-trained language model.

Given prompt representation $E_p$ and a set of prompt $pmi$, for different input $s_i$, we generate a scoring matrix $w \in R^{L \times L}$ to select the suitable template combination under specific semantics, this progress can be expressed by:

$$E_{p'} = \frac{exp\left(W s_i^\top + b\right)}{\sum_{j=1}^{M} exp(W s_i^\top + b)} E_p \tag{2}$$

where $E_p$ is the set of available prompts embeddings, $E_{p'} \in R^{M \times L \times H}$ is a weighted representation that blends the adapted prompt embeddings, incorporating both the input sentence and the entire adjusted prompt set's semantic information, and $W \in R^{L \times H}$ is a learned weight matrix that is used to adapt the selected prompt to match the semantics of the input sentence $s_i$.

The Hearst method (Hearst, 1992) is an efficient method for identifying hypernym (is-a) relation from a given sentence with exploiting certain lexico-syntactic patterns. To attenuate the impaction from the specific prompt, we use the Hearst pattern as a representation of knowledge. We give examples of these prompts in Table 2.

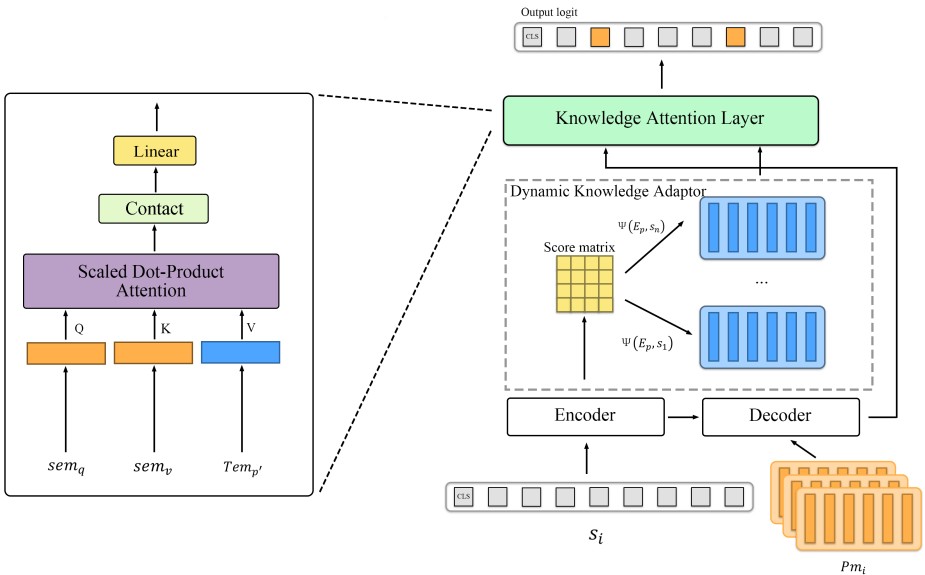

Figure 2: The overall framework of KEPL

Table 2: Examples of prompt used in KEPL

| Prompt |
| --- |
| 对象代表一个分类 |
| Object represents a classification |
| 对象表示一个分类 |
| The object signifies a classification |
| 对象同属一种分类 |
| Objects belong to the same classification |
| 对象是一种分类 |
| The object is a type of classification |

### 4.3 Unified representation for templates and context

To facilitate a more coherent integration of the structured prompts and unstructured text, we employ an attention mechanism, we learn a unified representation of context:

$$H_{s_i}^{(\mathcal{L})} = \mathcal{L}\left(s_i\right) \qquad (3)$$

where $H_{s_i}^{(\mathcal{L})} \in R^{L \times H}$ denotes the context representation. $\mathcal{L}$ now refers to the Language Model (LLM) and $H$ is the dimension of the contexts. This progression in the approach is leveraged to effectively integrate context and prompts.

To capture useful information and uncertainty from each view (knowledge and context), we learn a unified representation through specific attention. Specifically, as shown in Figure 2, we use $sem_q, sem_v, E_t$ to compute an attention output $x_{att} \in R^{L \times H}$. This process can be expressed as follows:

$$x_{att} = softmax\left(\frac{sem_q, sem_v{}^T}{\sqrt{d_k}}\right) Tem_{p'} \quad (4)$$

where $sem_q, sem_v$ are the linear projection of the $L(s_i)$, $Tem_{p'} \in R^{L \times H}$ represent the $E_{p'} \in R^{L \times H}$ through linear layer. This advancement is leveraged to facilitate enhanced interplay between templates and input information, promoting more effective integration of structured prompts and unstructured text.

### 4.4 Knowledge-guided extraction

In the process of knowledge-guided extraction, we aim to calculate the distribution of each token for the start or end of a given role based on the unified context-prompt representation. The logits reflect the unnormalized log probabilities of each token being the starting or ending positions of a target hypernym or hyponym. They are calculated via the linear projection from $x_{att}$ as follows:

$$\text{logits}_i^{start|end} = x_{att'}^{start|end} \cdot E_{p'}{}^T \qquad (5)$$

Here, $x_{att'}^{start|end}$ are the linear projections of $x_{att}$ that are computed separately for the start and end positions of a target feature (either hypernym or hyponym). $x_{att'} \in R^{1 \times H}$ represents the linear projection of $x_{att}$, $E_{p'}^T \in R^{H \times L}$, and $\text{logits}_i \in R^L$ symbolizes the contextual token distributions. Noted that different span $\text{Span}_{pi}$ will result in different corresponding $x_{att}$.

The logits are the scores assigned to each token in the input sentence and they measure the likelihood of each token being the start or end token of the span of interest. Once these logits are transformed into probabilities through a softmax operation, these values are utilized to determine the start and end positions for hypernym and hyponym spans in the text.

We observed sample imbalance in the extractive task, hence we utilize the focal loss function as follows:

$$\text{FL}(p_i) = -(1 - p_i)^\gamma \log(p_i) \quad (6)$$

$$p_i = \frac{logits_i}{\sum_{j=1}^M logits_j} \quad (7)$$

In these equations, $p_i$ represents the predicted probability, and to avoid the exhaustive threshold tuning, we use the softmax function to compute these probabilities.

The connotation for starting and ending positions of the target span $\widehat{s_k}$ is illustrated as :

$$(\widehat{s_k}) = set\ (i,j) \in \mathcal{C}\ \arg\max\ score_k\ (i,j) \quad (8)$$

The pair $(i, j)$ that maximizes $\text{score}_k(i, j)$, which gives us the span position of hypernym or hyponym in the sentence.

## 5 Experiment

In this section, we evaluate the efficacy of the proposed KEPL model. The evaluation experiments are presented in Section 5.1 and the results are discussed in Section 5.2. Furthermore, we assess the effect of varying the number of knowledge prompts in Section 5.3. An ablation study can be found in Section 5.4.

### 5.1 Evaluation and Settings

**Dataset** Our KEPL framework is further evaluated using the CHR Dataset, compiled from three diverse sources: We-Media, Baidu Encyclopedia, and various news outlets. This CHR dataset provides a more practical perspective on the applicability of our KEPL framework, as it encompasses a wide range of language styles and topics.

**Evaluation** Our experiments have been employed on CHR datasets. We briefly describe each below. In order to evaluate the performance of our experiments on CHR datasets, we employ precision, recall and F1 as our primary evaluation indicators.

**Baselines** We perform a comparison with the current state-of-the-art and also with some classical models to show the efficiency and effectiveness of the proposed KEPL model.

- W2NER(Li et al., 2022) which discontinuous NER (8 English and 6 Chinese datasets) top-performing baselines

- Ernie3(Sun et al., 2021) is a SOTA pretrained Chinese model outperforms the state-of-the-art models on 54 Chinese NLP tasks

- UIE(Lu et al., 2022) excels in Chinese information extraction tasks and is wildly used. .

**Implement Details** The CHR Dataset was employed in our experiments, divided into training, validation, and test sets in a 0.7:0.15:0.15 ratio. The model was trained with a batch size of 32 and a learning rate of 0.0001 using the Adam optimizer. The models were trained for five epochs, with early stopping implemented to prevent overfitting.

### 5.2 Experiment Results

We report the results of different methods in Table 3. A more visual presentation of the direct results from our experiment can be appreciated in Table 4. It can be seen that our method outperforms all other methods in F1 score and achieves a 2.3% improvement in F1 over the best method UIE (Lu et al., 2022). It shows the effectiveness of our proposed method. Furthermore, from Table 1, we also can see, unlike other methods that predominantly rely on localized features or word-level relationships, KEPL models effectively account for the broader semantic context. This approach enhances the accuracy of inferring hypernym relationships, even in the absence of explicit markers for hierarchical relationships.

Table 3: Different Performance of KEPL on Different Data Sources

| Method | Baike | | | We-Media | | | News | | |
|---|---|---|---|---|---|---|---|---|---|
| | R | P | F1 | R | P | F1 | R | P | F1 |
| W2NER | 0.9056 | 0.5655 | 0.8703 | 0.5028 | 0.5655 | 0.5323 | 0.7185 | 0.6841 | 0.7009 |
| Ernie3 | 0.9773 | 0.8356 | 0.9009 | 0.7926 | 0.3987 | 0.5306 | 0.9642 | 0.5844 | 0.7277 |
| UIE | 0.9769 | 0.8208 | 0.8921 | 0.7407 | 0.6348 | 0.7273 | 0.9574 | 0.6348 | 0.7634 |
| KEPL-Bart | 0.9785 | 0.8847 | 0.9293 | 0.7466 | **0.6728** | 0.7078 | 0.9067 | **0.8861** | 0.8294 |
| KEPL-Ernie3 | **0.9861** | **0.8861** | **0.9334** | **0.8613** | 0.6327 | **0.7295** | **0.9543** | 0.7717 | **0.8534** |

Table 4: Different model results for Chinese sentence

| Instance | 奥马哈海滩是第二次世界大战诺曼底登陆战役中, 四个主要登陆点之一的代号 |
|---|---|
| Chatgpt | < 奥马哈海滩，第二次世界大战诺曼底登陆战役 > 
 < 盟军四个主要登陆点之一, 第二次世界大战诺曼底登陆战役 > |
| Lexcial | < 奥马哈海滩, 世界大战诺曼底登陆战役 > |
| Pattern | < 奥马哈海滩, 第二次世界大战诺曼底登陆战役 > |
| Baseline model | < 奥马哈海滩, 主要登陆点 > |
| Our Method | < 奥马哈海滩, 代号 > |
| Adaptor prompt | 对象属于一种分类 |

### 5.2.1 Results on CHR Dataset

The KEPL-Bart and KEPL-Ernie3 methods achieved higher rates when compared to other models such as W2NER, Ernie3, and UIE. This demonstrates the efficacy of the KEPL approach, particularly the effectiveness of incorporating Hearst-like patterns as prompts and embedding patterns and text simultaneously.

The performance enhancements of our KEPL models are largely attributed to the implementation of the Dynamic Adaptor for Knowledge and knowledge attention modules, which effectively mediate the interaction between structured prompts and unstructured text, contributing to a superior hypernym-hyponym extraction.

Table 5: Metrics on the test set for the CHR

| | R | Micro F1 | F1 |
|---|---|---|---|
| W2NER | 0.5958 | 0.5488 | 0.6743 |
| Ernie3 | 0.8090 | 0.7164 | 0.7826 |
| UIE | 0.8846 | 0.7992 | 0.8432 |
| KEPL-Bart | 0.9138 | 0.8324 | 0.8665 |
| **KEPL-Ernie3** | **0.9413** | **0.8471** | **0.8701** |

The evaluation results of different methods are shown in Table 5, from which we have several observations:

1. Unstructured text lacks a predefined structure or explicit markers for hierarchical relationships, making it difficult to discern the underlying organization. Hierarchies are often implied through contextual cues, such as sentence structure, semantic relationships, or proximity of concepts, requiring sophisticated natural language processing techniques to identify and extract these relationships accurately.

2. The other methods are unable to recognize the<Hypernym, Hyponym> appears in pairs, they predominantly rely on localized features or word-level relationships, thereby disregarding the encompassing semantic context required for accurate inference of hypernym relations.

### 5.2.2 Results on specific scene

We compare KEPL performance with the baseline model on each scene in Table 3, observing the results on the Baidu, We-Media, and News datasets, the performance of the KEPL models varies.

Compared to other models, KEPL models show higher scores in almost all metrics across different datasets. This can be attributed to the Hearst pattern selector, which adapts to different sentence semantics, and the unified representation for templates and context, which effectively captures the

useful information from both the pattern and the text.

The results highlight the effectiveness of the KEPL approach in handling different text complexities and styles, thereby exhibiting promising potential for real-world hypernym-hyponym extraction tasks. Nevertheless, improvements could be made to better handle datasets with informal and context-specific language like the We-Media data.

### 5.3 Effect on Knowledge Number

In this section, we evaluate the effect of varying the number of knowledge prompts on the performance of our KEPL model in Table 6. We vary the number of prompts from 50 to 300 in increments of 50 and report the Recall (R), Precision (P), and F1 score in each case.

Table 6: Effect on Knowledge number for KEPL

| Prompt number | R | P | F1 |
|---|---|---|---|
| 50 | 0.8938 | 0.8210 | 0.8495 |
| 100 | 0.8999 | 0.8204 | 0.8523 |
| 200 | 0.9043 | **0.8293** | 0.8617 |
| **300** | **0.9138** | 0.8238 | **0.8665** |

As seen in Table 6, there is a clear improvement in the KEPL model's performance as the number of prompts increases. This confirms our hypothesis that increasing the volume of knowledge prompts can enhance the model's effectiveness.

### 5.4 Ablation Study

In this section, we further provide more insights with qualitative analysis and error analysisto address the remaining challenges. In Table 7, we display the results of individually removing components from the trained KEPL model on the CHR dataset.

In the *prompt w/o* experiment, we use a Random matrix to replace the Dynamic Adaptor, which leads to a drop of 6.73% in F1 score, showing the importance of using the prompt as the connection of hypernym and hyponym. The prompt instructs the model to consider the hierarchical structure between concepts and to accurately identify hyponyms and hypernyms. This approach provides a structured framework for the model to leverage contextual cues, linguistic patterns, and semantic associations related to hyponymy and hypernymy,

enabling it to capture and utilize the rich hierarchical information present in the text.

In the *attn w/o* experiment, we deliberately trained KEPL (Knowledge-Enhanced Prompt Learning) without knowledge-attention. This design choice resulted in a reduction in performance, primarily due to the model's compromised ability to effectively integrate context and prompts. Attention mechanisms play a crucial role in enabling the model to attend to relevant parts of the input and to appropriately align them with the given prompts. Therefore, the absence of attention mechanisms in our experiment negatively impacted the model's capacity to fully exploit contextual information and prompts, ultimately affecting its ability to accurately represent and extract hyponym-hypernym relationships from unstructured text.

In the *Knowledge-guided extraction w/o* experiment, we trained KEPL logits with a Linear MLP. We found that performance decreases by 4.97% in F1 score. This highlights the importance of incorporating prompts before directly merging them with the context, particularly in tasks related to acquiring hypernyms and hyponyms. The prompt serves as a guidance signal, providing explicit instructions to the model and aiding it in understanding the desired relationship or task. By incorporating prompts, the model gains a clearer direction and context-specific cues, which are crucial for accurately capturing and representing hierarchical relationships.

## 6 Conclusion

In this paper, we introduce Knowledge Enhanced Prompt Learning (KEPL) for extracting hypernym-hyponym relations in Chinese language. KEPL utilizes the concept of prompt learning to incorporate prior knowledge in the form of patterns into the model, which simultaneously embeds both the pattern and text.

The prompt in the framework uses Hearst-like patterns, specifically for extracting hypernym-hyponym relations. Additionally, we have created a Chinese hypernym-hyponym relation extraction dataset, which includes three different types of scenarios: Wikipedia, news articles, and We-media. The results of our experiments using this dataset show that our proposed model is both efficient and effective.

Table 7: Ablation Study with A for Konwledge attention, D for Dynamic Adaptor, E for Knowledge-guided extraction

|  | R | P | F1 |
|---|---|---|---|
| KEPL **w/o A** | 0.8667 | 0.8013 | 0.8328 |
| KEPL **w/o D** | 0.8201 | 0.7965 | 0.8082 |
| KEPL **w/o E** | 0.8295 | 0.8174 | 0.8234 |
| **KEPL** | **0.9138** | **0.8238** | **0.8665** |

## Limitations

**Domain Adaptability** While the KEPL model has demonstrated effective performance in scenarios such as Baike, News, and We-media, its applicability in other domains or contexts is yet to be confirmed. For example, the model may require further tuning and optimization for specialized domains such as technology, law, or medicine.

**Data Dependency** The performance of the KEPL model is to a significant extent dependent on the quality and quantity of available data. In cases where data is scarce, particularly in certain domains or for specific tasks, the model may require larger datasets for efficient training.

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
