# OpenReview forum: "KEPL: Knowledge Enhanced Prompt Learning for Chinese Hypernym-Hyponym Extraction"
_EMNLP/2023/Conference — EMNLP 2023 Main_

### Official Review · Reviewer_3dDj · 2023-07-22

**Soundness:** 4

**Excitement:**

4: Strong: This paper deepens the understanding of some phenomenon or lowers the barriers to an existing research direction.

**Missing References:**

A Short Survey on Taxonomy Learning from Text Corpora: Issues, Resources and Recent Advances. EMNLP 2017: 1190-1203

More than just Frequency? Demasking Unsupervised Hypernymy Prediction Methods. ACL/IJCNLP (Findings) 2021: 186-192

BiRRE: Learning Bidirectional Residual Relation Embeddings for Supervised Hypernymy Detection. ACL 2020: 3630-3640

Hypernymy Detection for Low-Resource Languages via Meta Learning. ACL 2020: 3651-3656

**Paper Topic And Main Contributions:**

This paper presents a  Knowledge Enhanced Prompt Learning for Chinese hypernym-hyponym extraction.

**Questions For The Authors:**

Table 2 and Table 4: hard to understand without English translations.

The presentation quality should be improved by proofreading.

Figure 1 is hard to read.

**Reasons To Accept:**

1. Using prompt learning for Chinese hypernym-hyponym extraction is novel.

2. Good performance compared to previous methods.

3. Good experimental analysis.

**Reasons To Reject:**

1. Missing references.

2. Lack of English translations in case study.

**Reproducibility:**

3: Could reproduce the results with some difficulty. The settings of parameters are underspecified or subjectively determined; the training/evaluation data are not widely available.

**Reviewer Confidence:**

5: Positive that my evaluation is correct. I read the paper very carefully and I am very familiar with related work.

---

> ### Author Rebuttal · Authors · 2023-08-29
>
> We are sincerely thankful for providing us with constructive feedback and detailed suggestions about our paper! We would like to address your feedback in the following ways:
>
> - **Missing references:** We appreciate the reviewer's suggestions on the missing references and will update our bibliography to include the recommended papers[1][2][3][4].
>
> - **Lack of English translations in case study (Table 2 and Table 4):** We apologize for this oversight and will provide English translations of the examples presented in Table 2 and Table 4 to ensure the content is clear and accessible to all readers.We agree with your suggestion about lacking of English translations.
>
> - **Presentation quality and proofreading:** We acknowledge that the presentation quality of the paper can be improved. We will thoroughly proofread the manuscript, address any grammatical errors, and ensure that the language is clear and concise. In addition, we will enhance the readability of Figure 1 to ensure that the information presented is easy to understand.
>
> - **Domain adaptability:** The reviewer raises a valid concern regarding the domain adaptability of our KEPL model. We recognize that the model has demonstrated effective performance in certain domains, but its applicability in other domains is yet to be confirmed. In future work, we will explore the extension of our model to other specialized domains, such as technology, law, or medicine.
>
> - **Data dependency:** We acknowledge that our KEPL model's performance heavily relies on the quality and quantity of the available data. In cases where data is scarce, particularly for certain domains or specific tasks, we will work on improving the model by creating or collecting larger datasets for efficient training.Furthermore, we are also committed to making our dataset open-source, and we plan to release it in the near future for the benefit of the research community.
>
> We affirm our commitment to rectify the errors you outlined in our revised manuscript. Your constructive feedback is deeply appreciated and we are grateful for your consideration of our work.
>
> Sincerely,
>
> The Authors
>
> [1]A Short Survey on Taxonomy Learning from Text Corpora: Issues, Resources and Recent Advances. EMNLP 2017: 1190-1203
>
> [2]More than just Frequency? Demasking Unsupervised Hypernymy Prediction Methods. ACL/IJCNLP (Findings) 2021: 186-192
>
> [3]BiRRE: Learning Bidirectional Residual Relation Embeddings for Supervised Hypernymy Detection. ACL 2020: 3630-3640
>
> [4]Hypernymy Detection for Low-Resource Languages via Meta Learning. ACL 2020: 3651-3656

---

### Official Review · Reviewer_rkKb · 2023-07-25

**Soundness:** 3

**Excitement:**

3: Ambivalent: It has merits (e.g., it reports state-of-the-art results, the idea is nice), but there are key weaknesses (e.g., it describes incremental work), and it can significantly benefit from another round of revision. However, I won't object to accepting it if my co-reviewers champion it.

**Missing References:**

[1] also builds a Chinese hypernym dataset. You should compare your dataset with theirs and explain why you want to build another one.

[2] Liu, Ming, Yaojia LV, Jingrun Zhang, Ruiji Fu, and Bing Qin. "BigCilin: An Automatic Chinese Open-domain Knowledge Graph with Fine-grained Hypernym-Hyponym Relations." arXiv preprint arXiv:2211.03612 (2022).

**Paper Topic And Main Contributions:**

The authors build a Chinese hypernym detection dataset by first using a rule-based approach to extract the candidate hypernym-hyponym pairs and next ask humans to verify. Then, the authors propose a new architecture to improve its performance on the proposed dataset over several baselines, including W2NER (state-of-the-art Chinese NER), Ernie3 (state-of-the-art pretrained LM), and UIE (widely used Chinese information extraction tool).

**Reasons To Accept:**

1. It constructs a new hypernym dataset in Chinese, which covers several data sources.
2. Some baselines are appropriate and it has some ablation studies.

**Reasons To Reject:**

1. One main problem of the dataset is that the authors use the rule-based approach to identify potential hypernym-hyponym pairs. That is, the dataset does not contain the hypernym-hyponym pairs that cannot be captured by the rules (by the way, what rules you use?). The examples in the paper in line 247-248) and Table 4 share a pattern. Thus, the dataset would favor the model that learns those patterns and I believe you can improve the performance on this dataset by simply using the same rule-based approach you used plus a trained model to verify the candidates.

2. It is unclear why we need to have a special architecture in Figure 2 for the hypernym detection task. If prompts are effective, couldn't we just use the typical prompt approach such as append XXX is a <mask> after the GPT or RoBERTa? You should compare your methods with these baselines to justify your special architecture design. There are many existing hypernym detection methods such as Dash et al., 2020 and Roller et al., 2018. You should also compare your methods with them. It's also good to see how well ChatGPT could do in this task.

**Reproducibility:**

3: Could reproduce the results with some difficulty. The settings of parameters are underspecified or subjectively determined; the training/evaluation data are not widely available.

**Reviewer Confidence:**

3: Pretty sure, but there's a chance I missed something. Although I have a good feel for this area in general, I did not carefully check the paper's details, e.g., the math, experimental design, or novelty.

**Typos Grammar Style And Presentation Improvements:**

The data construction process could be more clear. Who label the hypernym? What's the ratio of positive and negative pairs among those candidates from rule-based approach. What approach do you use for disambiguation and what do you mean the most appropriate pair?

Typos:
line 66: sub-entities
line 244 and 265: space after ,
line 318 and 286: the uppercase rule should be the same in your titles
line 329: pic 2 -> Figure 2
line 390, 393, 396: space before (
line 398: wildly used -> widely used
Dash 2020a is the same as Dash 2020b

---

> ### Author Rebuttal · Authors · 2023-08-29
>
> We value your insightful comments and thank you for raising these concerns. We would like to address your feedback in the following ways:
>
> - **Hypernym-Hyponym Pairs Extraction:** We appreciate the reviewer's comments and would like to clarify a potential misunderstanding regarding the content. It should ne noticed that KEPL dataset does contain many instances like (053-054) that cannot be captured by rule-based approaches, which is one of the major reasons we deem it necessary to use multi-source datasets. The sentences that pass these filters are then manually labeled with appropriate hypernym-hyponym relation by our team. Please refer our response to mrHG in **Construction of Dataset** for more detail.  We will provide a detailed section showing Construction of Dataset in the revised manuscript.
>
> - **Rule-based Approach as a Baseline:** Thank you for raising concerns about the rule-based method. It's worth noting that we did not follow a specific pattern in data prepossessing. The rule-based method does boast high precision rather than recall, as shown in the additional experiment. Indicative performance metrics for a rule-based approach are as follows：
>
> |         | Precision | Recall | F1 Score |
> |---------|-----------|--------|----------|
> | Pattern |    0.409  |  0.251 |   0.311  |
> | Lexical |    0.471  |  0.281 |   0.352  |
>
>
>  Our findings show that both precision and recall rates are relatively low when rule-based method applied to Chinese contexts.Noticed that the rule-based approach[1] used here in our experiment for comparison is different from the rule used during our dataset construction.
> - **Hypernym Detection Method:** Before clarifying the difference between our paper and the hypernym detection method, we should point out that we have compared with the recent SOTA methods to ensure the effectiveness and robustness of our proposed method in identifying hypernym-hyponym relationships in various contexts. The hypernym detection task is fundamentally interested in whether a given term pair "is" or "is not" in hypernymy relation. As a pipeline approach, Hypernym Detection could be prone to error propagation issues, supporting our assertions that a hypernym detection method may not be suitable baseline for this task[2].
>
> - **ChatGPT Evaluations:** We agree with your suggestion about including results from ChatGPT for a fair comparison. We are currently working on more elaborative tests in this regard and appreciate your patience. A crucial preliminary finding, as evidenced in Table 5, is the inability of ChatGPT to accurately acknowledge hypernymy relationships. For instance, in the example provided concerning Omaha Beach, one of the four primary landing points during World War II's Normandy Invasion, ChatGPT struggled to recognize "Omaha Beach" as a code name.
>
> - **Missing References:**  In comparing our work to "BigCilin", we believe that while it is a notable work in hypernym detection, it focuses on building a linguistic resource, the BigCilin, as an open-domain Chinese knowledge graph. In contrast, KEPL primarily targets the improvement of hypernym discovery (Given an input term, the hypernym discovery task retrieves a ranked list of suitable hypernyms from a large corpus) [2] by constructing a dataset specifically for this task.
>
> - **Data constructions:**
>
>   - We appreciate the input regarding the necessity to enhance the clarity of how the dataset was assembled. We hope this clarification proves helpful. We are grateful for your valuable feedback and will take it into account as we revise our manuscript.  We are also committed to making our dataset open-source, and we plan to release it in the near future for the benefit of the research community. Please refer to "Construction of Dataset" in response for mrHG for more detail.
>
>   - In order to establish a balanced representation, it is essential to maintain a 1:1 ratio of positive and negative pairs among the candidate sample as outlined in Table 1. This approach will counter any potential bias and provide a more comprehensive scope for rebuttal analysis.
>
>   - The meaning of "the most appropriate pair" is to select the most accurate or most appropriate combination of entities or elements in a specific context. In the example {Xie Houqin, a senior investment consultant of Jufeng Investment Advisory, told the Securities Daily that since July this year}, 'the most appropriate pairing' is "Xie Houqin, senior investment consultant of Jufeng Investment", not "Xie Houqin, senior investment consultant".
>
> We are committed to rectifying the typographical errors that you've pointed out in our revised manuscript. Your helpful critique is highly appreciated; thank you for giving our work your attention. Once again, we're grateful for your insightful commentary. We're eager to tackle these issues as we refine our manuscript.
>
>
>
> Sincerely,
>
> The Authors
>
> [1]https://github.com/HillZhang1999/Chinese-Hypernym-Hyponym-Relation-Extraction
>
> [2]Bai Y, Zhang R, Kong F, et al. Hypernym discovery via a recurrent mapping model[C]//Findings of the Association for Computational Linguistics: ACL-IJCNLP 2021. 2021: 2912-2921.

---

### Official Review · Reviewer_mrHG · 2023-08-03

**Soundness:** 3

**Excitement:**

3: Ambivalent: It has merits (e.g., it reports state-of-the-art results, the idea is nice), but there are key weaknesses (e.g., it describes incremental work), and it can significantly benefit from another round of revision. However, I won't object to accepting it if my co-reviewers champion it.

**Paper Topic And Main Contributions:**

In this paper, the authors built a Chinese hypernym relation extraction dataset, proposed a framework that learns a unified representation of context through specific attention, and evaluated the effectiveness of the proposed framework on the dataset.

The dataset is to amplify the quality and diversity across various domains against existing resources and is built from Baidu Baike, We-Media, and several news sources.
The proposed framework can inject knowledge for a given sentence and calculate the distribution of each token selected as start/end for each hypernym and hyponym using the unified representation of templates and contexts.

The experimental results using the dataset showed that the F1-measure of the proposed framework is higher than the ones of other baselines.

**Questions For The Authors:**

- A: The reviewer needs to understand the explanation of the proposed framework.
Could the authors explain the framework, including the shape of tensors and parameters and a specific example?

- B: How many and what lexico-syntactic patterns were used to create the dataset?

- C: How were the 300 prompts made, and what are they like?

- D: Is the proposed framework based on the assumption that the data must contain hypernym-hyponym relation instances?
The reviewer thinks the prompts of this paper may harm precision if the data does not have hypernym-hyponym relation instances.

**Reasons To Accept:**

- A new dataset for Chinese hypernym-hyponym relation extraction was built.
- The experiment results showed the effectiveness of the proposed framework.

**Reasons To Reject:**

- The method of dataset construction could be more transparent, including the specific rules and annotation guidelines used to extract candidates.
- It is difficult for the reviewer to know the dataset format (data, span1, span2). What is the "latent position" that span represents?
- The reviewer needs help understanding the explanation in section 4.2.
  - What is knowledge Ψ, specifically?
  - What is L and M?
  - Where and how is the scoring matrix w used?
  - What are s_{i} and s_{i,j}? (What is the shape of s_{i} and s_{i,j}?)
  - The reviewer needs to learn the specific calculation of equation (2).
  - What is the shape of pmi? How to obtain pmi?
  - What is B^{HxH}?
- This method may need the distribution of the start and end for each hypernym and hyponym, but the reviewer does not know how to calculate it from equation (6).
- Where does the framework switch the corresponding parameter for each?
- The reviewer needs to learn the score_k of equation (9).

**Reproducibility:**

2: Would be hard pressed to reproduce the results. The contribution depends on data that are simply not available outside the author's institution or consortium; not enough details are provided.

**Reviewer Confidence:**

3: Pretty sure, but there's a chance I missed something. Although I have a good feel for this area in general, I did not carefully check the paper's details, e.g., the math, experimental design, or novelty.

**Typos Grammar Style And Presentation Improvements:**

In Table 1
Cntext  ->  Context

In section 5.4
Pattern-guided extraction -> Knowledge-guided extraction

---

> ### Author Rebuttal · Authors · 2023-08-29
>
> We thank you for your valuable feedback! We appreciate your time and effort in providing feedback. We realize there are some areas of confusion and we’d like to address your concerns as follows:
>
> - **Construction of Dataset**:
> We acknowledge the feedback regarding the need for greater transparency in dataset construction. We employ certain specific rules to filter short sentences. We will provide a detailed section showing Construction of Dataset in the revised manuscript.These rules include the following:
>   - We disregard any sentences with a character count less than a given threshold (10 characters).
>   - We eliminate sentences with an excess of punctuation marks (e.g., commas, colons, periods).
>   - We filter sentences with numerals or certain non-Chinese characters.
>   - The sentences that pass these filters are then manually labeled with appropriate hypernym-hyponym relation by our team. We mark the entity position: Determine the start (entity _start) and end (entity _end) positions of the answer in the text, ensuring the scope of the answer is accurate. If there is no explicit hypernym-hyponym entity, mark the corresponding position as NONE.
>
>
> - **Dataset Format**:
>  Here, the 'latent position' refers to the position of words in the sentence that the spans denote. For example, in a sentence "Apple is a type of fruit", {span1} = {0,1} would denote the position of "fruit" (hypernym), and {span2} = {6,7} would denote the position of "Apple" (hyponym).
>
> - **Explanation in Section 4.2**:
> The knowledge Ψ refers to the dynamic adaptor for knowledge. It is a component that dynamically selects and applies the most appropriate prompts (patterns) based on sentence semantics. $L$ is the LLM and $M$ represents the number of prompt candidates available. The scoring matrix $w$ is used to generate a set of weights for each pattern, which indicates its relevance to the given sentence. Each sentence $s_i$ is an input of context which might contain hypernym-hyponym relation(like 'Apple is a type of fruit'). Regarding your question on the function $L_{dec}(pmi)$, it is used to generate prompt embeddings $E_p$, which have a shape of $N\times L\times H$ where $N$ is the number of prompts, $L$ is the length of prompts, and $H$ is the dimension of the contexts.
> For the parameter $pmi$, each component corresponds to a prompt and its shape is $N\times L$, where $N$ is the number of prompts and $L$ is their length. We apologize for the typographical error where the bias vector, '$b$', was mistakenly denoted as '$B$'. We assure you that it will be corrected in the revised manuscript.
>
> - **Parameters in Equation (6)**:
> The equation (6) does not directly calculate the distribution for start/end of hypernym and hyponym. It calculates the logits (unnormalized log probabilities) for each word in a sentence to be part of a relation. By applying softmax to these logits, we can obtain the probabilities which can then be used to infer start/end positions for hypernym and hyponym spans.
>
> - **Switching Corresponding Parameters**:
> Regarding the process of switching corresponding parameters for each within the proposed framework, it is achieved through the attention mechanism. Depending on the attention score, the model learns to focus more on certain words that are likely to be involved in the hypernym-hyponym relation.
>
> - **Equation (9)**:
> The score_k represents the "strength" of a pair (i, j) being the target span (hypernym and hyponym). It is obtained by running the unified representation for templates and context through several Transformer layers followed by the application of a softmax function.
>
> We hope this clarification resolves your concerns. As for the specific points:
>
> - **A.Explain framework**:
> Thank you for your advice. We agree that more illustrative examples would be beneficial. We will provide a detailed example showing the shape of tensors and parameters in the revised manuscript.
>
> - **B.Number of lexico-syntactic patterns**:
> For the question about the lexico-syntactic patterns used in building the dataset (question B), we did not rely on specific patterns to avoid potential biases. Our construction process involved manual annotation of the sentences that passed our filters, marking appropriate hypernym-hyponym relationships. This in turn helps our Knowledge Enhanced Prompt Learning (KEPL) model train more accurately on a broad range of hypernym-hyponym relations, rather than those limited to specific patterns only.
>
> - **C.Creation of 300 prompts**:
> The 300 prompts were manually created by our team. We constructed phrases that capture the 'is-a' relationship between two entities in diverse ways, an example can be seen in Table 2 of our paper. The 300 prompts were manually created by our team. We formulated phrases that encompass the 'is-a' relationship between two entities in a variety of ways, as demonstrated in Table 2 of our paper. It's important to note that these prompts were used only for knowledge purposes and were not utilized in the data construction process, as mentioned in question B.
>
> - **D.Assumption and potential precision harm**:
> While our method does aim to extract hypernym-hyponym relations, it is not reliant on the assumption that each input sentence must contain such relations. When a sentence does not contain targeted relations, our model can predict a null output.
>
> We will ensure that the typographical errors you highlighted will be corrected in our revised manuscript. We greatly appreciate your constructive feedback and thank you for considering our work. Thank you once again for your valuable feedback. We look forward to addressing these points as we revise our manuscript.
>
> Sincerely,
>
> The Authors

---

### Meta-Review · Area_Chair_wi6R · 2023-09-15

**Recommendation:** 4

**Metareview:**

The paper presents a new approach for hypernym discovery in Chinese and a new dataset for evaluation.

Reviewers mrHG and rkKb call out the dataset as an important contribution and all three reviewers acknowledge the performance improvements that the proposed method achieves.

The majority of issues raised by the reviewers come down to missing details in the manuscript, and other presentation issues. From the extensive and constructive discussions (especially with Reviewers mrHG and rkKb) it appears that the authors were able to address all these concerns, either with additional experiments or with clarifications. An additional issue raised by Reviewer rkKb regarding a missing simpler baseline was similarly addressed by the authors.

While the revisions required by the authors would be relatively extensive, the underlying soundness of the work is high, as evidenced by the increased scores by the reviewers.

---

### Decision · Program_Chairs · 2023-10-07

**Decision:**

Accept-Main

**Comment:**

The paper presents a new approach for hypernym discovery in Chinese and a new dataset for evaluation.

Reviewers mrHG and rkKb call out the dataset as an important contribution and all three reviewers acknowledge the performance improvements that the proposed method achieves.

The majority of issues raised by the reviewers come down to missing details in the manuscript, and other presentation issues. From the extensive and constructive discussions (especially with Reviewers mrHG and rkKb) it appears that the authors were able to address all these concerns, either with additional experiments or with clarifications. An additional issue raised by Reviewer rkKb regarding a missing simpler baseline was similarly addressed by the authors.

While the revisions required by the authors would be relatively extensive, the underlying soundness of the work is high, as evidenced by the increased scores by the reviewers.